# A Survey of Autoencoder Algorithms to Pave the Diagnosis of Rare Diseases

**DOI:** 10.3390/ijms221910891

**Published:** 2021-10-08

**Authors:** David Pratella, Samira Ait-El-Mkadem Saadi, Sylvie Bannwarth, Véronique Paquis-Fluckinger, Silvia Bottini

**Affiliations:** 1Center of Modeling, Simulation and Interactions, Université Côte d’Azur, 06200 Nice, France; david.pratella@univ-cotedazur.fr; 2Centre Hospitalier Universitaire (CHU) de Nice, Institute for Research on Cancer and Aging, Nice (IRCAN), Université Côte d’Azur, Inserm U1081, CNRS UMR 7284, 06200 Nice, France; saadi.s@chu-nice.fr (S.A.-E.-M.S.); bannwarthsylvie@yahoo.fr (S.B.); veronique.paquis@univ-cotedazur.fr (V.P.-F.)

**Keywords:** rare diseases, autoencoders, artificial intelligence, personalized medicine

## Abstract

Rare diseases (RDs) concern a broad range of disorders and can result from various origins. For a long time, the scientific community was unaware of RDs. Impressive progress has already been made for certain RDs; however, due to the lack of sufficient knowledge, many patients are not diagnosed. Nowadays, the advances in high-throughput sequencing technologies such as whole genome sequencing, single-cell and others, have boosted the understanding of RDs. To extract biological meaning using the data generated by these methods, different analysis techniques have been proposed, including machine learning algorithms. These methods have recently proven to be valuable in the medical field. Among such approaches, unsupervised learning methods via neural networks including autoencoders (AEs) or variational autoencoders (VAEs) have shown promising performances with applications on various type of data and in different contexts, from cancer to healthy patient tissues. In this review, we discuss how AEs and VAEs have been used in biomedical settings. Specifically, we discuss their current applications and the improvements achieved in diagnostic and survival of patients. We focus on the applications in the field of RDs, and we discuss how the employment of AEs and VAEs would enhance RD understanding and diagnosis.

## 1. Introduction

Genome regulation encompasses all facets of gene expression, from biochemical modifications of DNA to the physical arrangement of chromosomes and the activity of transcription mechanisms. Recently, several techniques have been developed to interrogate these complex processes in multiple dimensions (DNA, RNA, proteins, lipids, metabolites…), known as “omics”. While these approaches can reveal physio-pathological mechanisms in the sample, the joint use of several omics on the same sample is key in the understanding of the associated phenotype [1].

The limited number of samples that can be collected are usually noisy, incompletely annotated, sparse, and high-dimensional (many variables), making it very challenging to develop integrative computational approaches with regard to this type of data. Nowadays, several machine learning approaches have been proposed to analyze multi-omics datasets [2]. Specifically, unsupervised approaches learn representations by identifying patterns in the data and extracting meaningful knowledge, while overcoming data complexities. Among such approaches, unsupervised learning methods via neural networks such as autoencoders (AEs) or variational autoencoders [3,4] (VAEs) have shown promising performances [5], with applications on various types of data, such as single-cell data [6], multi-omics data [7], and metagenomics data [8] and in different contexts, such as cancer [9], bacterial infection [10] or in healthy patient tissues [11]. An AE learns a compressed representation (embedding) of the input data, passing the information through layers smaller than the previous one. The latent space will end up with a bottleneck layer composed of the most informative features of the original input data, and then will be used to reconstruct data in the most similar way. Through this process of compression, the algorithm will capture a better representation of the data structure (i.e., intrinsic relationships between the data variables), and therefore will allow for more accurate downstream analyses [12]. In this review, we will discuss about their usage in the field of rare diseases (RDs) and beyond, with a focus on why their implementation in such a context would be suitable in the near future.

### 1.1. RDs and Their Diagnosis

RDs are any disease that affects a small percentage of the population. In Europe, they affect less than 1 in 2000 citizens. There are more than 7000 RDs worldwide. Although individually rare, collectively, RDs are estimated to affect 350 million people globally. Most rare diseases are genetic and are present throughout a person’s entire life, even if symptoms do not immediately appear. RDs are characterized by a wide diversity of symptoms, which can vary from patient to patient and can also appear to be similar to those of common diseases. These factors imply that RDs can often be misdiagnosed. According to the Global Genes organization, 8 out of 10 RDs are caused by a faulty gene, and approximately 75% affect children, yet it takes an average of 4.8 years to arrive at an accurate diagnosis. This is part of the reason for 30% of children with RDs not living to see their fifth birthday. There are numerous challenges and issues that need to be addressed, ranging from technical to theoretical aspects, such as the small number of patients, often children, the heterogeneity of the disease, and the limited amount of national/international data resources [13,14,15].

The development of new technologies, such as genomic analysis by means of next generation sequencing (NGS) and other omics technologies, has boosted the molecular understanding and diagnosis of RDs [16,17,18,19,20,21,22,23,24].

Despite a significant leap in the diagnostics of rare genetic diseases in recent years, more than half of patients with a suspected RD remain without a definite diagnosis [25]. Patients with RD who are not diagnosed or diagnosed late may experience a delay in the start of a specific treatment, which, in turn, could have irreversible consequences for their health, may prevent informed reproductive choice and could cause great stress for patients and their families.

### 1.2. Omics and Multi-Omics Approaches for RD Diagnosis

The development of high-throughput technologies in the past decade allowed us to generate a large amount of different data type, each of them representing different levels of information ranging from DNA level to protein level, including data such as genome, proteome, transcriptome, epigenome, metabolome [26,27]. All of these multi-omics data attempt to capture the biological machinery occurring in the living being, providing a high level of information. However, each technology individually cannot depict the entire biological complexity of most human diseases. The combination of multiple data types can compensate for missing or unreliable information in any single data type, and multiple sources of different biological measurements could point to the same results and low down the number of false positive [28].

The current challenge is to integrate these data together in order to decipher new levels of information that could be key in RD diagnosis, by identifying the causal mechanisms of those diseases. AEs and VAEs are very promising technologies to integrate and analyze data from different sources (e.g., multi-omics, patient registries, …) that can be used to overcome further challenges, such as low diagnostic rates, a reduced number of patients, and geographical dispersion.

## 2. Artificial Intelligence Methods for Biology

For several years, the various techniques of machine learning and deep learning have been widely used in image and visual recognition problems. The models developed so far can be classified into three popular categories, which are supervised, unsupervised and semi-supervised learning. For supervised methods, algorithms are fed with “labeled” data during a training step to infer a function which then will classify data or predict outcomes [29]. The purpose of unsupervised learning is to identify structures and features from a training dataset without the use of labeled data [30]. Most known unsupervised methods concern clustering algorithms such as hierarchical clustering or k-means clustering and dimensionality reduction methods such as principal component analysis (PCA), t-distributed stochastic neighbor embedding (t-SNE) [31] and uniform manifold approximation and projection (UMAP) [32]. Among unsupervised methods, artificial neural networks (ANN), particularly AEs and VAEs, have emerged as very promising methods to work with various biological problems and to integrate diverse types of data. Finally, semi-supervised learning is a learning problem that involves a small number of labeled examples and a large number of unlabeled examples. Learning problems of this type are challenging, as neither supervised nor unsupervised learning algorithms are able to make effective use of the mixtures of labeled and unlabeled data. As such, specialized semi-supervised learning algorithms are required. Although very promising, semi-supervised learning methods have mainly been applied for medical image analysis [33], which is out of the scope of the present review.

### Basis of the AE Algorithm and Its Variant

AEs are composed of two main parts, which consist of an encoder and a decoder (Figure 1A). The encoder maps the highly dimensional input data into a latent variable consisting of one or multiple hidden layers of lower dimension. This bottleneck layer forces a compressed representation of the input data. The second part consists of the decoder, which attempts to reconstruct the input data from the embedding. The dimensionality reduction followed by the reconstruction of the input forces the model to only retain features with high variability, setting aside features with less variability. Autoencoders are often associated with the denoising procedure, because unimportant variations are automatically left out [34]. This loss is modeled through a loss function that considers the distance between compressed data and reconstructed data. The most commonly employed loss functions are mean squared error and Kullback–Leibler divergence.

Several variants of AEs have been proposed since they were first introduced. These variants mainly aim to address shortcomings, such as improved generalization, disentanglement, and modification to sequence input models. Some significant examples include the denoising autoencoder (DAE) [35] (Figure 1B), the sparse autoencoder (SAE) [36,37] (Figure 1C), and more recently the VAE [3,4] (Figure 1D). Each of these different generative models have their own specificity. The DAE takes as its input corrupted data for which some values have been randomly turned to zero. Usually, 50% of input nodes are set to zero; however, a lower percentage, around 30%, has been proposed [38]. This kind of AE has mainly been applied to images [35]. The SAE allows one to obtain a bottleneck layer without reducing the number of nodes in the hidden layers [36,37]. The loss function is defined using a sparsity penalty. This sparsity penalty can be defined by using the Kullback–Leibler divergence, which is a standard measure of the difference between two functions [36,37]. VAEs are probabilistic generative models, considering specific assumptions about the distribution of hidden layer features. They learn the true distribution of input features from latent variable distribution using the Bayesian approach and use stochastic inference to approximate a latent space defined by a mean “μ” and a standard deviation “σ” (Figure 1D). This property enables the possibility to compute probability distributions and thus generate new data [3,4]. The main difference between classical AE and VAE resides in the latent space which is continuous for the latter. These algorithms are scalable to large datasets and can deal with intractable posterior distributions by fitting an approximate inference or recognition model, using a reparametrized variational lower bound estimator. They have been broadly tested and used for data compression or dimensionality reduction [11,39,40,41,42,43,44]. Their adaptability and potential to handle non-linear behavior have made them particularly well suited to work with complex data [7,9,45,46,47,48]. A recent benchmark proposed VAEs as the best-performing methods to detect cancer subtypes compared with other type of AE [7].

## 3. AE Applications in Biological and Medical Contexts beyond RD

The first applications of AE on biological data date from 2016. Tan et al. [49] developed ADAGE, a DAE (Figure 1B) to study microbe–host interactions. They showed that ADAGE is able to identify biological patterns and to extract meaningful features. By comparing their method with PCA and independent component analysis, they demonstrated that ADAGE was better at regrouping replicate samples and the biological features extracted by ADAGE were not clearly captured by other methods. They then improved it by constructing ensemble ADAGE [50] (eADAGE) by combining many individual ADAGE models into a single model. For each eADAGE model, they combined 100 models with identical parameters but distinct random seeds. Wang et al. [51] re-used the ADAGE package [49] to create a DAE model and used it on transcriptomic data from patients with lung cancer. They identified a signature composed of 35 genes and concluded by proposing this signature as a novel diagnostic and prognostic biomarker for human lung adenocarcinoma. Chen et al. [52] used a SAE (Figure 1C) to study the transcriptomic machinery of yeast. This algorithm identified transcription factors with a fundamental role in yeast machinery by studying microarray gene expression. Furthermore, they found that SAE hidden layers correspond to common biological processes.

One example of the very first application of AE in the medical field is DeepPatient [53]. Taking advantage of electronic health records, Miotto et al. [53] developed a DAE-based method to improve clinical prediction for severe diabetes, schizophrenia and several cancers.

Finally, VAEs (Figure 1D) have been used in different biological contexts with different purposes, applied on different data type including proteomics, bulk RNA-seq and/or single-cell RNA-seq (scRNA-seq) and more. The origin of the data can vary, coming from healthy or diseased patients. Applications of AEs or VAEs on these data have been shown to improve downstream analysis and results, mainly for the identification of cell subtype, drug response prediction or multi-omics data integration.

Hereafter, we discuss some of the major applications of AE and its variants, in the biomedical field with an eye toward the advances in data analysis and developed algorithms (Appendix A, Figure 2).

### 3.1. AE Applications in Single Cell

Single-cell RNA sequencing (scRNA-seq) enables measurements of gene expression at the cell level and thus each of these cells will have its own transcriptome [54]. scRNA-seq allows to get a whole new level of information with more precision by comparison with bulk RNA-seq, where the sequencing results from a mixed cell population [55]. One of the main issues related to scRNA-seq data is the experimental noise that accompanies their generation. Indeed, at the single cell level, there is more variability in gene expression compared to an average cell population. Moreover, the low number of RNA transcripts available in single cell experiments will increase the rate of technical dropout events. This will provoke the scRNA-seq data to be highly sparse by including excessive zero counts that will cause the data to be zero-inflated, ending up with capturing only a small fraction of each cell transcriptome [56,57]. Recent research demonstrated the importance of correcting technical variation and showed improvement in downstream analysis [56,57,58]. One way to deal with this problem is to clean the data using a denoising algorithm [59]. Different works suggested several methods by using either a classical AE [34] or a VAE [6,11,39,40,60,61,62]. One solution is to go through an a priori modeling process. With VASC [39], the authors proposed to explicitly model the dropout events, which will help to find the non-linear hierarchical features representation of the original data. Using this approach, the authors asserted that VASC provided better dimension reduction and variational inference (scVI), both used a zero inflated negative binomial (ZINB) to model scRNA-seq noise. The ZINB allows one to take into account the RNA-seq count distribution, the overdispersion and the sparsity of the data by modeling the noise distribution in highly sparse count data. This causes the tool to learn gene-specific parameters such as the mean, dispersion and dropout probability and showed improvement in differential expression analysis, increasement in protein and RNA co-expression, enabling the discovery of subtle cellular phenotypes and increasing the correlation structure of key regulators [63].

In addition to their denoising purpose, used for correcting the batch effect or dropout events, AEs can also be applied to single cell data for other tasks. For example, scGen [6] enables the prediction of events caused by an external perturbation due, for instance, to drugs or infection. Based on the association of a VAE and vector arithmetic, it models perturbation and infection response of cells across different cell types, studies and species. It works by learning cell-type and species-specific responses from features that distinguish responding from non-responding genes and cells. To demonstrate the performance of their tool, they applied it to the human Peripheral Blood Mononuclear Cells (PBMC) dataset [64] stimulated with interferon (IFN-β), showing good prediction on gene expression for stimulated CD4-T. They also evaluated scGen on data from Haber et al.’s [65] study, consisting of two datasets of intestinal epithelial cells impacted by *Salmonella* or *Heligmosomoides polygyrus* infection.

Another example of the use of VAE in single cell data is scVAE [11] and SCA [66], which are employed for classification/clustering tasks. scVAE uses different types of VAE with either a Gaussian or a Gaussian-mixture latent variable prior. It is able to obtain a higher Rand index, an index measuring the similarity between different clusters, showing better performances than Seurat [67], the state-of-the-art analysis tool for single cell data. On the other hand, SCA uses a SAE and showed good results for clustering single cell data, highlighting functional features. For example, the authors were able to identify genes highly involved in monocytes functionalities.

Semi-Supervised Generative Autoencoder (SISUA) [68] is a semi-supervised model based on the association of a VAE and CITE-seq (Cellular Indexing of Transcriptome and Epitopes by Sequencing) data. CITE-seq is a technique allowing researchers to obtain information from surface proteins. Because of the low amount of dropout in CITE-seq [69] data, the authors took advantage of this property to improve SC clustering results and notably obtained better separation between CD8 and CD4 proteins.

Recent technological advances have enabled simultaneous acquisitions of multiple omics data at the resolution of a single-cell, thus producing “multimodal” single-cell data. The first developed methods based on VAE for multi-omics analysis at single cell level were scMVAE [70] and totalVI [71]. These models have some limitations, including extensive pre-processing of data for training and latent variable interpretation difficulties. To overcome these limitations, Minoura et al. proposed scMM, a novel statistical framework for single-cell multi-omics analysis specialized in interpretable joint representation inference and predictions across modalities [72].

### 3.2. AE Applications in Cancer

Another application of AE on biological data concerns cancer data analyses. Several tools have been proposed with different strategies and different aims. Indeed, these methods focus either on drug response prediction [48,73,74] or on subtype cancer classification/stratification [7,47,51,73,75].

#### 3.2.1. Drug Response Prediction

DeepDR [74] combines a network approach with AE. It is composed of three networks: i) a mutation encoder, ii) an expression encoder and iii) a drug response predictor network. The researchers showed that their tool performed better in drug response prediction compared to linear regression and SVM. The application of DeepDR revealed novel resistance mechanisms and drug targets. With the same aim, DeepProfile [73] and Dr.VAE [48] employ a VAE configuration. While DeepProfile uses a pre-trained VAE combined with a separately trained linear model to predict drug response, Dr.VAE is a semi-supervised method that learns a latent embedding of the gene expression used to feed a logistic regression classifier. The training data result from the combination of all the microarray datasets of the GEO database [76] for acute myeloid leukemia. Most of these methods outperformed currently used methods such as linear regression, SVM, PCA, k-means clustering [77].

#### 3.2.2. Cancer Classification and Stratification

Cancer classification and stratification are fundamental to adapting the treatment depending on the cancer subtype and/or the prognostic, since cancer stage is closely related to cancer survival [78]. To address these tasks, different tools have been proposed. Tybalt et al. [75] proposed a VAE-based method learning features recapitulating tissues specific patterns. By training it on the cancer genome atlas (TCGA) dataset [79], the authors were able to identify different features such as patient sex, allowing classification, to compare melanoma tumors to other cancer type and to identify high-grade serous ovarian cancer (HGSC) subtypes. The stacked sparse auto-encoder (SSAE) is a semi-supervised deep learning strategy for cancer prediction using RNA-seq data [80]. This approach outperformed other methods for all three cancer data sets tested in various metrics.

Zhang et al. [47] proposed multi-omics data integration with an AE associated with K-means clustering to stratify high-risk neuroblastoma. They showed that their AE algorithm outperforms other non-AE methods such as iCluster [81] and PCA. Another method for multi-omics integration for cancer classification is OmiEmbed [82]. It combines the basic structure of VAE with a classifier to perform task-oriented feature extraction and multi-class classification. It yielded better performances than methods using only one type of omics data. With the same aim, Hira et al. [83] proposed the Maximum Mean Discrepancy VAE (MMD-VAE), which outperformed multi-omics analysis of ovarian cancer data.

To improve genomic functional characterization, Chen et al. [84] developed a gene superset autoencoder (GSAE), a multi-layer autoencoder model with the incorporation of a priori defined gene sets. They introduced the concept of the gene superset, an unbiased combination of gene sets, with weights trained by the AE, where each node in the latent layer is termed a superset, with the goal of determining the functional or clinical relevance of the learned gene supersets from the model.

Franco et al. [7] benchmarked four types of AE algorithms, including classic AE, DAE, SAE and VAE, to identify subtypes of cancer among glioblastoma multiforme, colon adenocarcinoma, kidney renal clear cell carcinoma and breast invasive carcinoma. They showed that even though AE performances varied depending on the dataset used, classical AE and VAE showed the best results, performing better than standard techniques for dimensionality reduction such as PCA, kernel PCA, and sparse PCA.

### 3.3. VAEs Structure for Data Integration

One application of VAE concerns multi-data integration [8,9,85], which is currently a challenge of high interest in computational biology. Several methods and configurations already exist [86], but without any clear consensus of the best one to use. Simidjievski et al. [9] proposed four different architectures for data integration using VAE structures: Variational Autoencoder with Concatenated Inputs (CNC-VAE), X-shaped Variational Autoencoder (X-VAE), Mixed-Modal Variational Autoencoder (MM-VAE) and Hierarchical Variational Autoencoder (H-VAE). By comparing the performances of these different methods, they showed that the H-VAE and X-VAE outperformed the other configurations, with a more stable behavior for the first one. The authors also suggested that data integration performances rely on data types, with some types being more amenable.

The study of Nissen et al. [8] proposed VAMB, a VAE to integrate two distinct data types. The first type is the sequence abundance defined by the individual number of reads mapped to each sequence. The second is the k-mer distribution, which corresponds to the number of substrings of length *k* contained in a sequence. By using these two types of data, they outperformed existing state-of-the-art tools. Another tool for data integration is deepDR [85], a network-based approach for studying drug repositioning where the authors integrated 10 different networks, including drug–disease, drug–side-effect, drug–target and drug–drug networks. By converting topological structure of each network into vector representation by using a random walk with restart algorithm, the authors were able to construct a positive point-wise mutual information (PPMI) matrix then fed to multimodal deep autoencoder (MDA) to concatenate all the different network. They extracted the low-dimensional features from the middle layer of the MDA and then used it in a collective VAE (cVAE) to predict potential associations between drugs and diseases. By comparing their methods with baseline methods including random forest, kernelized Bayesian matrix factorization, support vector machine, random walk restart, they obtained better results.

One remarkable example of multi-omics data integration with AE is Multiview Factorization AutoEncoder (MAE) [87]. It combines multi-view learning, matrix factorization and AE with biological knowledge to integrate multi-omics data such as gene expression, DNA methylation and miRNA expression. The important contribution of this work is the introduction of external domain knowledge such as biological interaction networks to improve model generalizability and reduce the risk of overfitting. Another example of multi-omics integration through AE is Multiple Similarity Network Embedding (MSNE) [88]. MSNE integrates the multi-omics information by embedding the neighbor relations of samples defined by the random walk on multiple similarity networks. MSNE achieved outstanding performances for cancer subtyping compared to five other non-AE-based multi-omics integrative methods.

### 3.4. AE Applications in RDs

The development of omics technologies significantly improved the diagnosis of RDs. However, their success rate for detecting the responsible gene is far from complete. To fill this gap, the employment of RNA sequencing has been proposed as a complementary assay [89,90,91,92,93]. However, classical statistical methods have limited applications in the context of RD [94]. Consequently, there is an urgent need to develop novel computational approaches to resolve diagnostic deadlock and improve our knowledge of RD [1]. Brechtmann et al. developed OUTRIDER [95], an algorithm that uses an AE to model read-count expectations according to the gene covariation resulting from technical, environmental, or common genetic variations. The tool takes advantage of the generative model algorithm which reconstructing the RNA-seq data by fitting a negative binomial distribution and then computing a p-value and a Z-score. They used the Genotype Tissues Expression (GTEx) database [96], in which they injected simulated outliers in order to assess the sensitivity and the specificity of their tool. Additionally, the authors used data from Kremer et al.’s [89] study, consisting of individuals affected by rare mitochondrial disorders, with the goal to retrieve the aberrant gene expression manually identified and experimentally validated in the original publication.

Aberrant splicing is a major cause of rare disease. It is estimated that splicing mutations are responsible for 15–60% of human disease mutations [97,98,99]. By proposing FRASER [100], Mertes et al. responded to the lack of statistical significance assessments for splicing events in the field of RD. Their tool is based on a DAE and takes advantages of a beta binomial distribution, which takes overdispersion into account, and therefore is more suited for splicing events. To evaluate their tool, the authors used the same strategy employed for OUTRIDER. They injected splicing outliers in the GTEx and Kremer et al. datasets, and then computed a two-sided p-value along with a Z-score. To correct for multiple testing genome-wide, they used the FDR. FRASER showed better results compared to other methods, allowing them to identify several alternative splicing events including intron retention.

Although only few applications of AE and VAE has been developed in the context of RD, they have proven to be effective and have improved the diagnosis of RD. Thus, we foresee a rise of the employment of these technique in the field of RD.

## 4. Discussion and Conclusions: Open Challenges and Future Directions

With the recent advances in omics data production, we are able to perform various analyses. Omics enable us to enlarge the scope of biological data employed, enriching analysis and results. This progress has reduced the number of patients in diagnostic impasse, but it is still not enough. Multi-omics approaches are very promising for improving diagnostic performances, but several problems remain to be solved. Data analysis methods for multi-omics are generally developed for cancer research, where large numbers of samples are available, which is not the case for RD. Therefore, there is a need to develop multi-omics approaches applicable to small cohorts. In this review, we comprehensively collected the basic but essential concepts and methods of AE, together with its recent applications in diverse biomedical studies (Figure 2). We have showed that the use of machine learning methods such as AE or VAE algorithms can improve analysis and results. However, few methods have been applied yet to RD. The identification of pathogenic events through measurement of aberrant gene expression levels is a very promising approach. With their tool based on an AE algorithm, Brechtmann et al. [95] successfully identified pathogenic genes candidates; however, the use of negative binomial distribution to model RNA-seq data limits the employability of the tool. To date, no methods have been proposed for multi-omics analysis in the field of RD. One of the challenges is the limited number of samples for RD with respect to other pathologies. The limited number of patients is not the only difficulty in applying existing algorithms for multi-omics integration to RD. These diseases are rare and heterogeneous, and the causative gene(s) are usually unique or “private” for each patient (or family). They require a methodology that identifies unique signatures, making it difficult to apply most of the multi-omics methods available because they are more suitable for identifying common signatures. AE proved to be useful in multi-omics data integration and could open the way to better-performing methods, especially in RD; however, they have some weaknesses [101]. They are highly sensitive to parameter tuning. It has been pointed out how the performances of each method could vary upon those hyperparameters. In their review, Hu and Green [101] proposed relying on independent third parties to benchmark and assess the different methods and tools. Extensive benchmarks are needed to learn more about AE and VAE performances in RD.

Despite the fact that the implementation of AE and VAE algorithms in RD is still in its infancy, it has opened the door to a more faithful understanding of the complex aspects of RD physiology, pathology, and treatment. Although much remains to be learned and developed, we believe that this review has captured the essence of this field and will enhance the use of AE in RD and inspire future breakthroughs in both the understanding and diagnosis of RD.

## Figures and Tables

**Figure 1 ijms-22-10891-f001:**
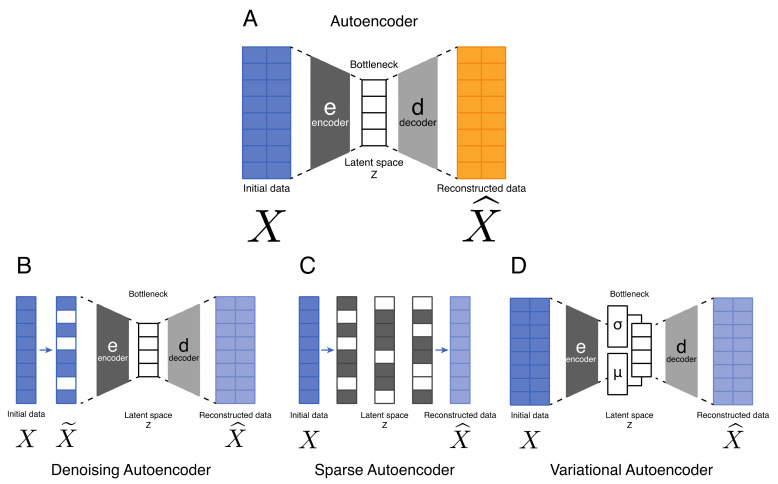
Different types of autoencoder. (**A**)—The classical autoencoder (AE) is composed of two main parts. First the encoder, annotated as “e”, encodes the input data through a latent space “Z” by reducing the data dimensionality. The latent space corresponds to a vectorial space with a bottleneck constraint in order to force the algorithm to keep only the most variable features. The second part corresponds to the decoder, annotated as “d”, that reconstructs the input data using the features encoded in the latent space. (**B**)—Denoising autoencoders (DAEs) are a category of AE where the input data are corrupted by setting nodes to a value of 0 (indicated in white). (**C**)—The sparse autoencoder (SAE) uses a penalty function. By penalizing the use of certain nodes (grey), these nodes are inactivated (white). Thus, the network is forced to learn features without reducing the number of nodes. (**D**)–The peculiarity of a variational autoencoder (VAE) is that the algorithm learns a distribution from the latent space “Z”. This distribution is defined by a mean “μ” and a standard deviation “σ”.

**Figure 2 ijms-22-10891-f002:**
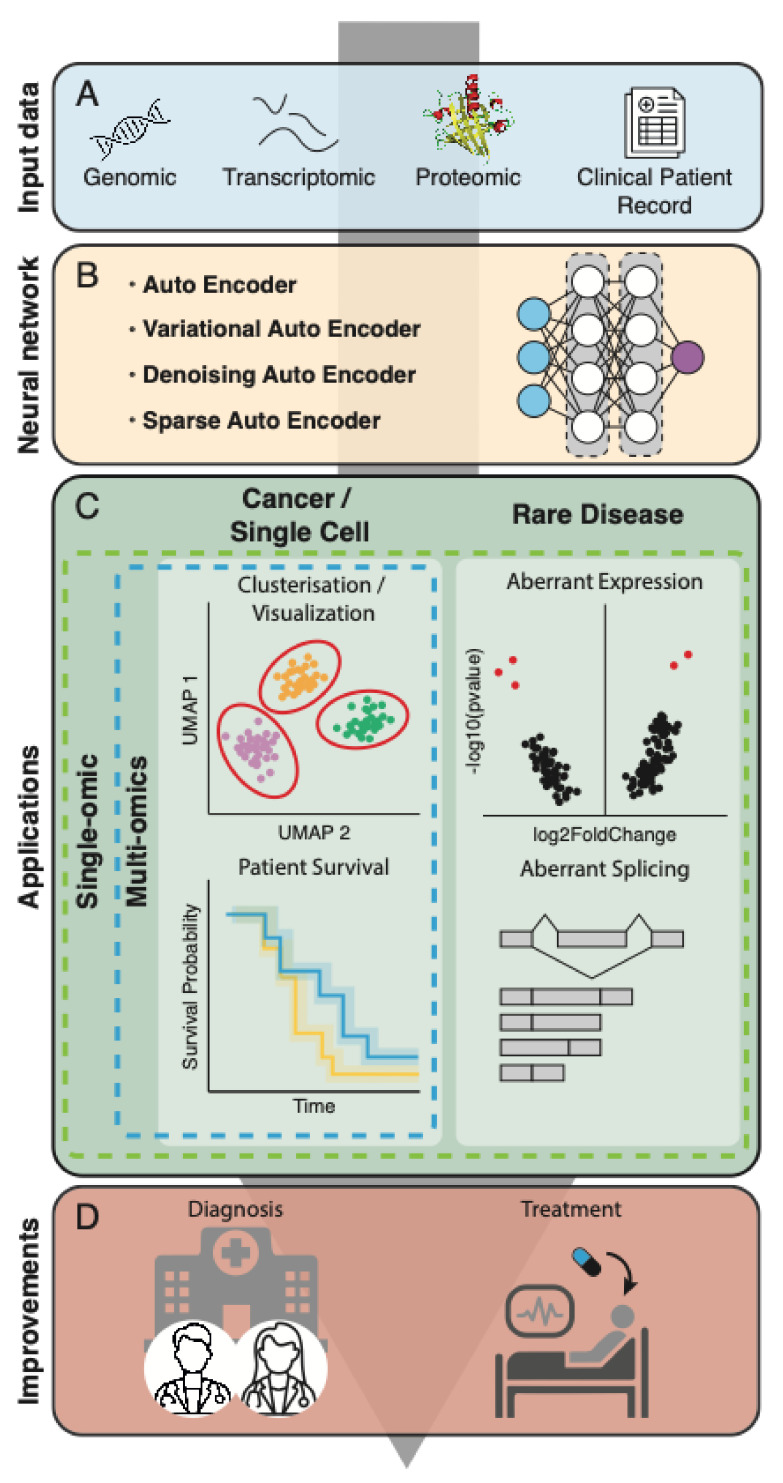
Autoencoders for personalized medicine approaches. (**A**)—Input omics data that can be of different types, such as omics, genomics, transcriptomics, proteomics, or clinal patient records. (**B**)—An AE-like algorithm is fed by input data as a single data type (single-omic) or multi data (multi-omics). (**C**)—Most common applications of these algorithms in the biomedical field and their achievements in terms of data analysis. Green dotted line groups AE applications on single-omic data, whereas blue dotted lines on multi-omics data. (**D**)—The results of the previous steps will enable improving patient diagnosis and treatment by providing powerful bioinformatics tools to physicians.

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
