# Peer review of "A Survey of Autoencoder Algorithms to Pave the Diagnosis of Rare Diseases"

_ijms, 2021, doi:10.3390/ijms221910891_

Round 1

Reviewer 1 Report

The review paper discusses the various applications of AE in biomedical studies and intends to propose the potential use of AE in the aspect of diagnosis and treatment of rare diseases. Overall, this will be a valuable review article for the scientific community. Here are my suggestions for the authors to make this paper even better.

Major issues:

  • As the title suggested, "Rare diseases" (RD) should be the main focus of this review article; however, the section "AE applications in RD" is relatively short in the main text. I know the field is still at its infancy stage, but the authors can provide their perspectives on how to use AE algorithms in the RD research.
  • Table 1 is not so useful unless it can also provide the original biomedical context used in each study. For example, the XXX tool has been used to solve the problem of HNSCC classification based on ssRNA-seq data.

Minor issues:

  • Line 284: By example -> For example
  • Line 357: To “fill” this gap...
  • Line 360: There is “an” urgent burning
  • Figure 3: Low resolution

Author Response

« The review paper discusses the various applications of AE in biomedical studies and intends to propose the potential use of AE in the aspect of diagnosis and treatment of rare diseases. Overall, this will be a valuable review article for the scientific community. Here are my suggestions for the authors to make this paper even better. »

Major issues:

« As the title suggested, "Rare diseases" (RD) should be the main focus of this review article; however, the section "AE applications in RD" is relatively short in the main text. I know the field is still at its infancy stage, but the authors can provide their perspectives on how to use AE algorithms in the RD research. »

  • We agree with the reviewer’s suggestion and we now discuss further about this topic, please see lines 664-682.

« Table 1 is not so useful unless it can also provide the original biomedical context used in each study. For example, the XXX tool has been used to solve the problem of HNSCC classification based on ssRNA-seq data. »

  • We thank the reviewer for this comment. We have now added the column “Purposes” to provide the biomedical context of the AE applications reported. We moved the table to Supplementary Materials to have more space to add all these additional details.

Minor issues:

« Line 284: By example -> For example

Line 357: To “fill” this gap...

Line 360: There is “an” urgent burning

Figure 3: Low resolution »

  • We apologize for the typos and mistakes, we have now revised the manuscripts and corrected accordingly. We also tried to provide a better resolution for Figure 3 (Figure 2 in the revised manuscript) but when we load in Word, the quality get worst. We will provide the high quality resolution one as additional file.

Reviewer 2 Report

This is a useful review paper that, in this reviewer's opinion, would be of interest to the broad rare disease (RD) research community. The review briefly introduces autoencoder (AE) methodology, comprehensively summarizes recent AE applications in the biological spaces, and then pivots to the (limited) existing applications in the RD context, concluding that many more such applications will, or should, be forthcoming.

The manuscript organization, however, is not optimal and can be improved:

Figure 1 is confusing (for example, the upper right quadrant should not be labeled "testing") and unnecessary --- a brief explanation of supervised/unsupervised learning in the text should be sufficient.

Most of the examples in section 2 should really go into section 3, as they are biology-related. Section two should concentrate on the mixed data types integration in the broader context. 

The authors cite many examples of successful AE applications in the biomedical space and conclude that, more often than not, they are "better" than the more conventional approaches. Just how better? Better generalization classification accuracy? Better at the specific predictors' discovery? Better at dealing with noise? Each example should, ideally, be concluded with a brief explanation of the "added value", if any. 

Figure 3 is, in this reviewer's opinion, trivial and unnecessary. It would be more useful to specify the biological application areas (when present) in Table 1.

An in-depth explanation of semi-supervised learning is missing. This is an important and promising research area. 

Finally, some of the terminology is wrong, and the authors should double-check their description of the mathematical and computer science concepts throughout the manuscript. For example, "Kullback-Leibler divergence which is a standard function to measure the difference between two function" --- KL divergence is a measure, not a function. 

There are numerous grammar issues throughout the manuscript (for example, "two main parts which consist in [sic] an encoder and a decoder"; "autoencoder process are [sic] often associated to [sic]"). This needs to be addressed before publication. In its present form, the manuscript is readable, but occasional errors (e.g., "has emerged as a very prone [sic] method to work with various biological problems") might confuse the readers. 

Author Response

« This is a useful review paper that, in this reviewer's opinion, would be of interest to the broad rare disease (RD) research community. The review briefly introduces autoencoder (AE) methodology, comprehensively summarizes recent AE applications in the biological spaces, and then pivots to the (limited) existing applications in the RD context, concluding that many more such applications will, or should, be forthcoming.

The manuscript organization, however, is not optimal and can be improved:

Figure 1 is confusing (for example, the upper right quadrant should not be labeled "testing") and unnecessary --- a brief explanation of supervised/unsupervised learning in the text should be sufficient. »

  • We agree with the reviewer and we have now removed Figure 1.

« Most of the examples in section 2 should really go into section 3, as they are biology-related. Section two should concentrate on the mixed data types integration in the broader context.

  • We thank the reviewer for this remark and we have now moved paragraph 2.2 into section 3. It is now paragraph 3.3.

« The authors cite many examples of successful AE applications in the biomedical space and conclude that, more often than not, they are "better" than the more conventional approaches. Just how better? Better generalization classification accuracy? Better at the specific predictors' discovery? Better at dealing with noise? Each example should, ideally, be concluded with a brief explanation of the "added value", if any. »

  • We have revised Table 1 and added the column “Advantages” to explain, when reported in the original publication, the “added value” of AE algorithms respect to other conventional approaches. We moved the table to Supplementary Materials to have more space to add all these additional details.

« Figure 3 is, in this reviewer's opinion, trivial and unnecessary. It would be more useful to specify the biological application areas (when present) in Table 1. »

  • We have revised Table 1 and added the column “Purposes” to provide the biomedical context of the AE applications reported. We moved the table to Supplementary Materials to have more space to add all these additional details.

« An in-depth explanation of semi-supervised learning is missing. This is an important and promising research area. »

  • We agree with the reviewer and we have added semi-supervised learning explanation in paragraph 2.

« Finally, some of the terminology is wrong, and the authors should double-check their description of the mathematical and computer science concepts throughout the manuscript. For example, "Kullback-Leibler divergence which is a standard function to measure the difference between two function" --- KL divergence is a measure, not a function. »

  • We apologize for the mistake, we have now corrected and revised the terminology.

« There are numerous grammar issues throughout the manuscript (for example, "two main parts which consist in [sic] an encoder and a decoder"; "autoencoder process are [sic] often associated to [sic]"). This needs to be addressed before publication. In its present form, the manuscript is readable, but occasional errors (e.g., "has emerged as a very prone [sic] method to work with various biological problems") might confuse the readers. »

  • We apologize for the mistakes, we have now revised the grammar through the manuscript.

Reviewer 3 Report

In this review, the authors discuss how autoencoders  (AE) and variational autoencoders (VAE) are used in the diagnostic setting and for the prediction of survival of patients suffering from rare diseases. Overall, the field of multi-omics approaches in diagnosing various disease types is timely and very interesting. The authors nicely organize the different approaches (AE and VAE) and refer to important contributions in the field. English is fine.

Fig. 1: The subfigures A and B are exact copies. Thus, although explained in the Figure legend, the differences between supervised and unsupervised learning are not visually clear.

Author Response

« In this review, the authors discuss how autoencoders  (AE) and variational autoencoders (VAE) are used in the diagnostic setting and for the prediction of survival of patients suffering from rare diseases. Overall, the field of multi-omics approaches in diagnosing various disease types is timely and very interesting. The authors nicely organize the different approaches (AE and VAE) and refer to important contributions in the field. English is fine. »

  • We thank the reviewer for this appreciation.

« Fig. 1: The subfigures A and B are exact copies. Thus, although explained in the Figure legend, the differences between supervised and unsupervised learning are not visually clear. »

  • We agree with the reviewer and we have now removed Figure 1 as suggested by reviewer 1 as well.

Round 2

Reviewer 2 Report

The revised version of the manuscript is much improved (in particular, the supplemental table is much more useful now). This reviewer would still recommend additional proofreading, as the minor grammar, etc., imperfections remain, but otherwise, the manuscript is ready for publication.